# Intuitive or Dependent? Investigating LLMs' Robustness to Conflicting Prompts

## Abstract

This study investigates the robustness of Large Language Models (LLMs) when confronted with conflicting information between their internal memory and external prompts. Such conflicts are frequently encountered in real-world applications, notably in retrieval augmentation LLM-based products. Specifically, we establish an evaluation framework and focus on two types of robustness, factual robustness, from an external objective perspective, targeting models' performance to identify the correct fact from prompts or memory, and decision style, from an internal model perspective, analyzing LLMs' preference for prompt and memory, categorizing models as intuitive, dependent, or rational, drawing upon cognitive theory. Our findings, derived from extensive experiments on seven open-source and closed-source LLMs, reveal that these models are highly susceptible to misleading prompts, especially for instructing commonsense knowledge. While detailed instructions can mitigate the selection of misleading answers, they also increase the incidence of invalid responses. After unraveling the model's preference, we intervene with different-sized LLMs through the specific style of role instruction to change this preference, this step allows us to measure their adaptability in role-playing—a capability that, though crucial, had not been quantitatively assessed before. We also analyze the change in external performance after intervening in internal preference.

## 1 Introduction

Large language models (LLMs) have become fundamental tools and achieved great success in the area of natural language processing (Wei et al., 2022; Mirowski et al., 2023). They can solve various tasks in the same form of text generation simply by providing task-specific prompts (Mishra et al., 2022). However, LLMs sometimes fail to understand and follow the prompted instructions. Take the inverse scaling prize as an example, when the instruction goes against common sense or refines some fake facts, the performance dramatically decreases even with increasing model scale. One of the main reasons is that LLMs may struggle between the memory and the conflicting prompt (McKenzie et al., 2022). This uncertain behavior can lead to poor performance in scenarios such as retrieval augmentation, where prompts often involve conflicting information.

To address this challenge, we propose a systematic framework to quantify the robustness of LLMs in conflict situations. Our analysis encompasses two perspectives: an external, objective perspective that considers the correctness of the LLMs' responses (Factual robustness), and to delve deeper into the correctness, we adopt an internal model perspective that focuses on the LLMs' preferences (Decision style).

**Factual robustness** measures the performance of LLMs to discern the facts in conflicting situations. There are two scenarios. Firstly, the model memorizes the correct facts while the prompt introduces a fake one; and secondly, the

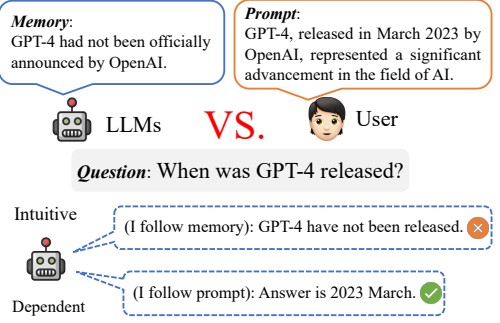

Figure 1: Under conflict intuitive models more rely on memory, while dependent models more rely on prompts to answer questions.

model's internal memory is inaccurate or lacks related knowledge, where the correct counterpart is provided in the prompt. Thus, if a model has a higher factual robustness score, it should be able to robustly ignore the prompted noisy information and better utilize the given external knowledge. Such a robust model is invaluable for fact-centric tasks like fact-checking or factual question-answering.

**Decision style** measures the LLM's internal preference. That is, regardless of the correctness of the answer, how LLMs make consistent choices — leaning towards the prompt or its own memory? (Refer to Figure 1 for an illustration of the two decision-making styles.). Assessing models' decision styles empowers users with insights into the models' behavioral inclinations. A higher score indicates that the model prefers memory and can yield less random answers, making it more predictable in non-factual applications, such as personalized assistance or recommendation.

To this end, we establish a complete benchmarking framework including a dataset, a robustness evaluation pipeline, and corresponding metrics. For the dataset, to ease the measurement and ensure the data quality, we leverage existing knowledge-intensive datasets and standardize a unified form of Multi-Choice Questions (MCQ). Under this setting, the "confliction" goes into where the knowledge presented in the prompt advocates for one answer, while the model memory suggests another one. For the evaluation pipeline and metrics, we design five steps from 1) memory assessment to 2) factual robustness in zero-shot and 3) few-shot in-context learning (ICL), to 4) decision style analysis, and finally 5) role play intervention as well as leaderboard building. To measure factual robustness, we break it into two aspects, Vulnerable Robustness (VR) and Resilient Robustness (RR), according to the two factual conflicting scenarios mentioned above. Toward Decision style, drawing from prior research (Harren, 1979; Phillips et al., 1984), we define three types of decision styles: intuitive, dependent, and rational, to categorize the models' behavior — to which extent they leverage internal memory or external prompt only, or can rationally consider both. Furthermore, to explore whether can change this inner preference, we implement interventions targeting the LLMs' inner preference for either memory or prompts by introducing specific role-based instructions. This advanced step enables us to measure models' adaptability in role-playing. This aspect of adaptability is particularly crucial, given the growing popularity of role play as a method to direct model behavior, yet it has not been quantitatively assessed before. By altering the models' internal preferences, we explore the effects on their external performance — factual robustness, thereby getting the upper bound of it.

We have conducted extensive experiments on seven closed-source and open-source LLMs. The main findings are as follows: **(1)** Compared with utilizing correct prompted knowledge, LLMs are more vulnerable to misleading prompts, thus enhancing VR robustness against noisy or fake prompts will be a pivotal focus in future research (Sec 4.1). **(2)** LLMs are more robust in using factual knowledge than commonsense knowledge via prompts. This suggests that we can leverage the retrieval-then-prompt strategy to remedy factual flaws while enhancing LLMs' inherent commonsense reasoning ability (Sec 4.1). **(3)** Detailed instructions are not magic. Although optimizing prompts with hints of possible noise does deter models from selecting misleading answers, the side effect leads to more invalid responses (Sec 4.2). **(4)** Medium-sized LLMs with instruction-tuning tend to exhibit a decision-making style dependent more on external prompts. Compared with them, GPT-4 and Bard are rational considering both memory and prompt. We attribute to the large model scale that amplifies memory retention while maintaining instruction-following capabilities (Sec 4.4). **(5)** We indeed can change LLMs' preference through role-playing intervention, while different LLMs vary a lot in adaptivity. Notably, although GPT-4 demonstrates the best performance and LLaMA2 is competitive in some aspects, the adaptivity reveals their large gap (Sec 4.5).

## 2   KRE DATASET CONSTRUCTION

To ensure data quality, our knowledge robustness evaluation dataset (KRE) extends existing machine reasoning comprehension (MRC) and commonsense reasoning (CR) datasets by automatically generating conflicting cases. We choose the tasks of wiki-based MRC and CR, as questions in wiki-based MRC are crafted based on Wikipedia, testing factual knowledge, and CR questions delve into commonsense knowledge. This combination is pertinent given that LLMs have demonstrated good memorization of factual and commonsense knowledge and can facilitate the robustness assessment. Specifically, each sample in our KRE dataset consists of four components: 1) question, 2) a set of answer choices, including an answer (**golden/correct answer**, $a_{gol}$) that conforms to facts or common sense and several misleading answers, 3) two types of contexts, a golden context to provide necessary facts or commonsense, and a negative context that supports a misleading answer (**negative answer**, $a_{neg}$), and 4) instructions. Since the questions, answers, and golden contexts are already

in MRC and CR datasets, we design three steps for KRE construction: dataset filtering, conflict generation, and instruction design. Note that our pipeline can be easily extended to a broader of tasks.

**Dataset filtering** For data sources, we select and process four publicly available datasets as the fundamental datasets to construct our KRE dataset: two MRC datasets MuSiQue (Trivedi et al., 2022) and SQuAD v2.0 (Rajpurkar et al., 2018), as well as two CR datasets ECQA (Aggarwal et al., 2021) and e-CARE (Du et al., 2022). We take the MRC paragraph and CR explanation as golden context for they are either based on Wikipedia or human knowledge. This setup enables us to verify the scope of LLMs' memory by withholding golden context. We only retain answerable examples for MRC and leverage the validation set. The KRE dataset comprises a total of 11,684 samples, more statistics results of the KRE are shown in Table 7.

**Conflict generation** This step involves the generation of misleading answer choices and negative context. As CR datasets have misleading choices, we utilize ChatGPT (OpenAI, 2022) to supplement MRC samples (Details can be found in Appendix B.1.1). Subsequently, we randomly choose one misleading option as the negative answer ($a_{neg}$) and employ ChatGPT to generate a negative context. Specifically, for SQuAD and MuSiQue, we substitute the golden answer entity in the gold context with the negative answer ( A case is shown in Appendix B.1.2 ). In the case of ECQA and e-CARE, we create an explanation tailored for the negative answer to serve as the negative context.

**Instruction design** Since the instruction in the prompt tells LLMs what to do, it may have some potential impact (positive or negative) on the usage of the knowledge in the prompts (Shi et al., 2023), leading to inaccurate robustness evaluation results. Hence, we propose and select different kinds of instructions to alleviate this potential problem. Based on how to use the knowledge in the context or few-shot examples, we design two kinds of instructions (1) **Instruction without hint** will not explicitly tell the LLMs how to use the knowledge or few-shot examples (if provided) to answer the question. (2) **Instruction with hint** tells LLMs there might be some noise in the knowledge context or few-shot examples (if provided), they should judge the quality of the prompts carefully. For each kind of instruction, We engaged four individuals to draft a total of 12 distinct instructions. After that, to further enhance the diversity of the instructions, we ask ChatGPT, GPT-4 (OpenAI, 2023), Claude (Anthropic, 2023), to rephrase the instruction, generating fresh variants. Consequently, we amassed a pool of 24 unique candidate instructions. Instructions are shown in Appendix B.2 and B.3.

**Human evaluation** We conduct the human evaluation for the quality of the negative context, involving four evaluators. We randomly select 100 questions from each corpus within the KRE dataset and provide the evaluators with the negative context, the associated question, and the set of answer choices. The evaluators are then tasked to assess the extent to which the negative context steers toward the negative answer option. The result shows that more than 98% of the sampled negative context is misleading. All the evaluation results can be referred to Appendix A.2.

## 3 METHOD

### 3.1 FRAMEWORK

**Preliminary:** Our evaluation focuses on the conflict situation where the prompt we consider has four key components: **the instruction** $I$, **the testing question** $x$, **the knowledge context** $C$ related to $x$, and **the few-shot examples** set $E$ (removed for zero-shot learning scenario). In specific, we introduce the knowledge context into example as $\hat{E}$. We define the prompt $P$ as the concatenation of the above components: $P = I \oplus E \oplus C \oplus x$, where $\oplus$ denotes the concatenation operation. For example, $P$ could be "**I:** *Help me to answer the question.* **E:** *Question: Where can I find water? Answer: Lakes.* **C:** *Foxes hunt chickens.* **x:** *Question: Where would I not want a fox?*".

**Framework:** The overall framework is shown in Figure 2. The entire pipeline consists of 5 steps: (1) **Memory Assessment** (Sec 3.2) to check if LLMs memorize the accurate knowledge to the question, partitioning the dataset for the following steps, (2) **Factual Robustness Evaluation** (Sec 3.3) targeting factual discernment in two conflict scenarios, vulnerable and resilient robustness, by supplementing with either negative or golden context according to different memory assessment results, (3) **Influence of Few-shot Example** (Sec 3.4) that further considers the impacts of the noise in few-shot examples on the robustness, complementary to the above zero-shot settings, and (4) **Decision-Making Style Analysis** (Sec 3.5) to reveal LLMs' preference between memory and prompt under conflict and categorize models as intuitive, dependent, or rational. (5) **Role Play Intervention and Leaderboard** (Sec 3.6) Upon evaluating the models, we construct a leaderboard

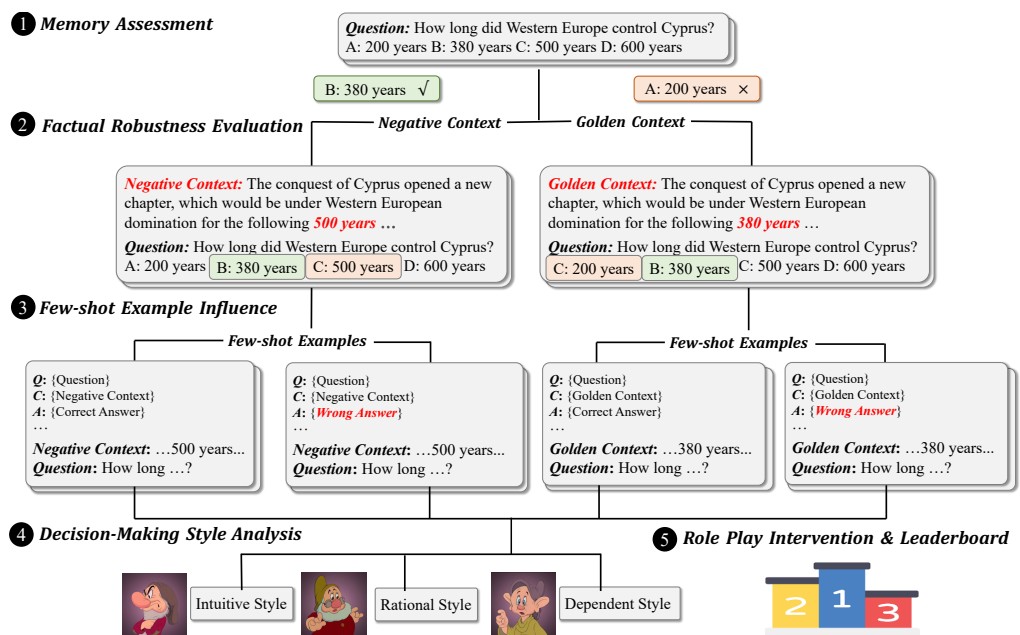

Figure 2: The overall robustness evaluation framework incorporates three key steps to assess the performance and robustness of LLMs: 1. Memory Assessment in Section 3.2. 2. Factual Robustness Evaluation in Section 3.3. 3. Few-shot Example Influence in Section 3.4. 4. Decision-Making Style Analysis in Sec 3.5. 5. Role Play Intervention and Leaderboard in Sec 3.6

based on the factual robustness result. Additionally, we implement role play intervention to control the preference of the model and discern the adaptability of the model.

## 3.2 MEMORY ASSESSMENT

Assessing memory in LLMs can be approached through two primary methods: **Direct Knowledge Evaluation** which entails directly evaluating the LLM on text that is a part of its training data. While this approach is direct and efficient, it is contingent upon access to the precise training datasets. **Question-Answering Assessment** employs question-answering tasks to gauge if the LLM has the knowledge necessary for accurate response generation. This strategy's primary appeal is that it eliminates the need for access to pre-training data, enabling the creation of a unified evaluation framework suitable for an extensive array of both open-source and closed-source LLMs. Based on these reasons we deploy the Question-Answering Assessment approach. We prompt the LLM question directly to conduct a memory assessment, and based on the answer we split the datasets into two parts: $D^+$ and $D^-$ for each LLM. Here, $D^+$ signifies those samples where the LLM's predictions are accurate. On the other hand, $D^-$ represents the samples where the LLM either provides incorrect answers or outputs that are invalid ( like " None "). That is, we assume that it possesses the corresponding memory to answer the question, if the model is able to answer the question correctly in this setting, because there is no other information (i.e., knowledge context).

## 3.3 FACUTAL ROBUSTNESS EVALUATION

Given $D^+$ and $D^-$ from Section 3.2, we supplement each sample input with extra negative or gold knowledge context to evaluate Factual Robustness. In these scenarios, We break down the overall factual robustness into two aspects: 1) **Vulnerable Robustness (VR)** that measures to which extent the model can trust its own correct memory even with a misleading prompt, and 2) **Resilient Robustness (RR)** to quantify the model's ability to harness accurate information from the prompt, when memory is insufficient or flawed. Formally, for each sample in $D^+$, we change the prompt to $P = I \oplus C^- \oplus x$, marked as $(D^+, C^-)$, to perform VR evaluation. Conversely, for each sample in $D^-$, we change the prompt to $P = I \oplus C^+ \oplus x$, marked as $(D^-, C^+)$, to measure the RR. We define the robustness metrics for VR and RR as follows:

$$VR_{(D^+,C^-)} = \frac{1}{|D^+|} \sum_{x \in D^+} \mathbb{I}\left[f(x, c^-; M) = a_{gol}\right], \; RR_{(D^-,C^+)} = \frac{1}{|D^-|} \sum_{x \in D^-} \mathbb{I}\left[f(x, c^+; M) = a_{gol}\right].$$

(1)

Here function $f(x, c; M)$ signifies the answer choice produced by model $M$ for the question $x$ with the provided context $c$, $a_{gol}$ is the golden answer (Defined in Sec 2) for question $x$. Notice that all the VR scores and RR scores are between 0 and 1. The greater VR score shows better robustness in maintaining the correct knowledge, and the greater RR score shows better robustness in using correct knowledge. On the other hand, a VR score nearing 0 indicates a stronger inclination of the model to rely on the external negative context. Similarly, a RR score closer to 0 reveals a model's preference to trust its internal memory. Using these two scores together, we represent the Factual Robustness:

$$FR = Avg\left(VR, RR\right) \tag{2}$$

Before assessing the robustness, we undertook an instruction selection process to mitigate the potential biases introduced by specific instruction. We conduct preliminary experiments on each LLM using a smaller sampled KRE dataset to identify the most effective instruction (Construct in Sec 2). Based on the result, we chose the instruction that exhibited the highest robustness for the Factual Robustness assessment. This selection process is also conducted for few-shot setting.

### 3.4 FEW-SHOT EXAMPLE INFLUENCE

To delve deeper into the effects of noise within few-shot examples on factual robustness, in addition to the previously explored zero-shot settings (Section 3.3), we introduce few-shot examples denoted as $\hat{E}$. Formally, the complete prompt is $P = I \oplus \hat{E} \oplus C \oplus x$. For Vulnerable Robustness, marked as $(D^+, C^-, \hat{E})$, and Resilient Robustness, symbolized as $(D^-, C^+, \hat{E})$. In specific, the few-shot example $\hat{E}$ testing for the VR is in the form $\hat{E} = C^- \oplus x \oplus A$, and when evaluating the RR the $\hat{E}$ is designed as $\hat{E} = C^+ \oplus x \oplus A$. In practice, the examples may also be noisy. We manually design gold and noisy examples that form the following three configurations: **All-positive** where few-shot examples correctly correspond to its question. This setting guides the model to rely on prompt when lacking correct knowledge and to overlook incorrect information when possessing the right knowledge. **All-negative** means the answer in each few-shot example is wrong to the corresponding question. This setting misleads the model to rely on the negative context and ignore the golden context. **Mixed** is a combination of positive and negative examples. In experiments, each of the above configurations shares the same questions. The examples are written by human annotators. We manually sample $m = 3$ samples for each evaluation setting. The corresponding VR and RR metrics under the few-shot setting are shown follow. $E_x$ is the few-shot examples configurations set (all-positive, all-negative, and mixed) corresponding to question $x$.

$$
\begin{aligned}
VR_{(D^+, C^-, \hat{E})} &= \sum_{x \in D^+} \frac{\sum_{e \in E_x} \mathbb{I}\left[f(x, c^-, e; M) = a_{gol}\right]}{|E_x||D^+|}, \\
RR_{(D^-, C^+, \hat{E})} &= \sum_{x \in D^-} \frac{\sum_{e \in E_x} \mathbb{I}\left[f(x, c^+, e; M) = a_{gol}\right]}{|E_x||D^-|}.
\end{aligned} \tag{3}
$$

### 3.5 DECISION-MAKING STYLE ANALYSIS

From the work work (Harren, 1979; Phillips et al., 1984) there are three kinds of decision-making styles: **Rational Style**: Rational decision-makers employ strategic approaches, taking into account both their personal preferences and external information to make informed decisions. **Dependent Style:** These decision-makers heavily rely on external information or the advice of others. **Intuitive Style**: This style of decision-makers is driven primarily by their inner feelings and instincts. Based on these decision-making features, we conceptualize a model's reliance on its internal memory for responses as learning from inherent instincts, and its deference to prompts as relying on external information sources. Building on this analogy, we defined a **Decision-Making Style Score (DMSS)** to measure the behavior of the LLM. With just one score, the DMSS, we can efficiently classify models into Rational, Dependent, or Intuitive categories.

$$
\begin{aligned}
DMSS = \frac{1}{|D|} &\left( \sum_{x \in D^+} \mathbb{I}\left[f(x, c^-; M) = a_{gol}\right] + \sum_{x \in D^-} \mathbb{I}\left[f(x, c^+; M) = f(x; M)\right] \right) \\
- \frac{1}{|D|} &\left( \sum_{x \in D^+} \mathbb{I}\left[f(x, c^-; M) = a_{neg}\right] + \sum_{x \in D^-} \mathbb{I}\left[f(x, c^+; M) = a_{gol}\right] \right),
\end{aligned} \tag{4}
$$

The closer DMSS to 1 means the model is more likely an intuitive decision-maker who depends on self-memory to answer the question. Conversely, when DMSS nearing -1 the model aligns more

with the dependent style, leaning heavily on external prompts. A score around 0 denotes a rational style, implying the LLM will consider the memory and the prompt together to make the decision. However, it's vital to note that a DMSS near 0 doesn't necessarily guarantee the model's capability to judiciously consider both the memory and the prompt. Given the conflicting scenarios in this study, discerning whether the model genuinely integrates both sources or randomly selects an option becomes challenging. Thus, in such cases, the Factual Robustness score should also be examined as an auxiliary metric to provide a more comprehensive understanding.

## 3.6 ROLE PLAY INTERVENTION

To further explore the modulation of decision-making tendencies of LLMs, we introduced a common intervention method "Role Play". This approach involves guiding the model's decision-making process through specifically designed role prompts. We designed two distinct role prompts to steer the models into specific decision-making pathways: **Dependent Role**: In this intervention, the model is furnished with a role prompt that asks it to prioritize information solely from the external prompt when generating answers. The aim is to see how a model behaves when explicitly told to disregard its internal knowledge and place full trust in the provided prompt. **Intuitive Role**: Contrary to the dependent role, this role prompt is designed to push the model towards relying predominantly on its intrinsic memory. The model is encouraged to harness its accumulated knowledge and insights, essentially making decisions that stem from its inner memory, irrespective of the external prompt (Prompt is shown in Appendix B.4). By adopting this approach, we can effectively measure the models' capability to exhibit dependent and intuitive behaviors. The resulting data, indicated by the lowest and highest DMSS, shed light on the models' **adaptability** in role-playing.

## 4 EXPERIMENT

We initially selected two LLMs ChatGPT and Vicuna-13B for experiments with the complete KRE dataset and analyze their behavior. Recognizing the importance of a broader analysis, we expanded our scope by incorporating five additional LLMs into our evaluation. However, due to computational constraints and the time-intensive nature of exhaustive tests, these models were assessed on a representative subset of the KRE dataset. Subsequently, we applied the role play intervention to these models and deployed the robustness leaderboard for comparisons

## 4.1 HOW FACTUAL ROBUST ARE LLMS ?

Following the framework, we conduct memory assessment. The overall memory assessment for Chat-GPT and Vicuna-13B is shown in Table 1. The result shows that the memory of ChatGPT possesses

greater and more accurate factual and commonsense knowledge than that of Vicuna-13B. Interestingly, both ChatGPT and Vicuna tend to perform better on commonsense knowledge datasets compared to factual ones. This might be because language models capture many co-occurrence relationships, and a lot of

| Model | ECQA$_{KRE}$ | e-CARE$_{KRE}$ | MuSiQue$_{KRE}$ | SQuAD$_{KRE}$ |
|---|---|---|---|---|
| ChatGPT | 74.2 | 81.5 | 34.6 | 65.3 |
| Vicuna-13B | 39.5 | 70.1 | 17.7 | 32.3 |

Table 1: The memory assessment results of ChatGPT and Vicuna-13B on the KRE dataset.

commonsense knowledge is an induction of these observed patterns. Subsequent to memory assessment, for every LLM we proceed with factual robustness evaluation. Prior to assessing the robustness of LLMs ChatGPT and Vicuna-13B, we conduct preliminary experiments on each LLM using a smaller sampled KRE dataset to identify the most effective instruction (Construct in Sec 2). Based on the results, we retain the top-performing instruction for both categories under evaluation. And use the best robustness score among these two as the final result. The selection result is shown in AppendixA.1. The factual robustness result is shown in Figure 3. ChatGPT and Vicuna exhibit similar behavior in terms of the two robustness. Specifically, A higher RR score relative to the VR score **indicates that LLMs already possess a stronger capability to utilize the correct knowledge from prompts. However, their robustness against negative context introduced by conflicting prompts remains suboptimal. Consequently, as the field progresses, enhancing robustness against negative context is likely to emerge as a paramount research focus.** Moreover, the observed $RR_{(D-,C+)}$ score on the two MRC datasets appears to0 be higher compared to the CR datasets, and the $VR_{(D+,C-)}$ scores are lower on the MRC portion of the KRE dataset. **This result shows that LLMs prioritize the prompts with factual knowledge more than with commonsense knowledge.** Consequently, when equipped with accurate internal knowledge, models are more

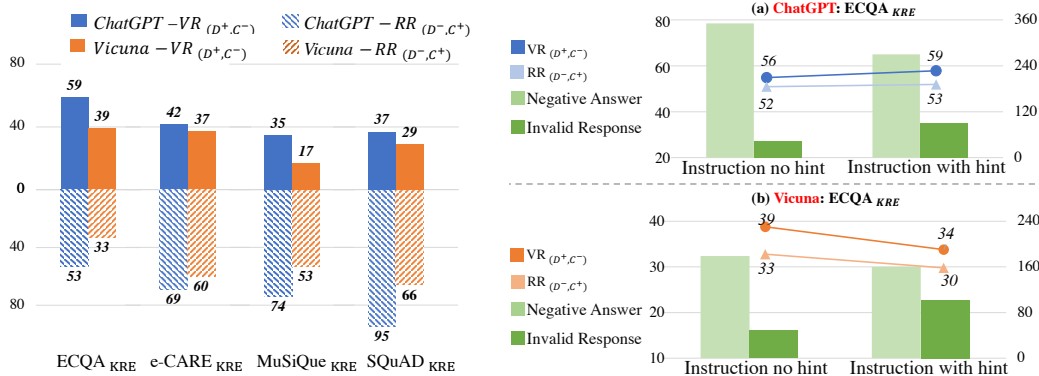

Figure 3: The Vulnerable Robustness score (%) and The Resilient Robustness score (%) for model ChatGPT and Vicuna-13B.

Figure 4: RR and VR of ChatGPT (a) and Vicuna (b) under different instruction settings: with and without hint (Sec 2). And the corresponding output number of Negative Answers and Invalid responses.

inclined to trust the prompts, leading to a decreased VR. Conversely, when their internal memory is lacking or incorrect, it results in an elevated RR. **Thus, to ensure better utilization of LLMs, there's a pressing need to enhance the precision of factual knowledge embedded in prompts. Meanwhile, when it comes to commonsense knowledge, the focus should be on amplifying the intrinsic memory of the model.** In the graph, the combined lengths of the bars representing VR and RR scores quantitatively illustrate the model's factual robustness. It's evident that ChatGPT's bar is longer than that of Vicuna-13, signifying that ChatGPT possesses superior factual robustness. This higher performance can be attributed to ChatGPT's larger number of parameters, more extensive training dataset, and enhanced instruction comprehension capabilities.

## 4.2 HOW DOES INSTRUCTION INFLUENCE FACTUAL ROBUSTNESS?

In our factual robustness evaluation process, we carefully select instructions as the preliminary study (defined in Sec 3.3), following the defined process. In this section, we explore the influence of different configurations of the instructions on the robustness. The results in Figure 4 (full results in Figure 8) indicate that neither ChatGPT nor Vicuna showcases any substantial improvements, though there is slight enhancement performance on the CR dataset for ChatGPT. This outcome seems counter-intuitive. To gain deeper insights, we further investigated the model's responses. Specifically, we calculate the number of negative answers and invalid outputs generated by the model. Our observations reveal that the **inclusion of a hint indeed reduces the propensity of the model to choose the negative answer. However, it also introduces an increase in the frequency of invalid responses,** especially for Vicuna. Therefore, when taking both factors into account, the overall robustness does not exhibit any marked improvement.

## 4.3 HOW DOES FEW-SHOT EXAMPLE EFFECT FACTUAL ROBUSTNESS?

Similar in zero-shot setting 3.3, before assessing the robustness, we select the best performance instruction. As for the instruction influence in the few-shot setting, we observe a phenomenon consistent with that in the zero-shot scenario. More in Figure 8. The robustness score for ChatGPT and Vicuna-13B under few-shot setting can be found in Figure 5. The results demonstrate that for both ChatGPT and Vicuna, the "All-positive" configuration exhibits the highest RR and the highest VR. However, when compared to the zero-shot setting ($VR_{(D^+,C^-)}$ and $RR_{(D^-,C^+)}$) "All-positive" setting do not always have a positive effect under the conflict situation. This phenomenon is counter-intuitive, conventionally, one would anticipate the "All-positive" approach to augment perfor-

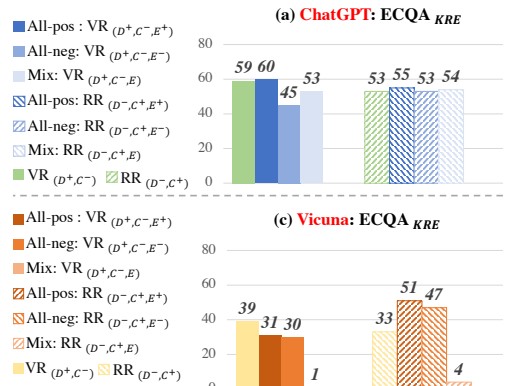

Figure 5: The VR and RR score (%) under the influence of three few-shot configurations.

mance, "All-negative" to impede it, and "Mixed" to lie somewhere in between. **The result indicates that the few-shot approach doesn't consistently bolster performance, even in an "All-positive" versus zero-shot comparison.** Two potential explanations emerge for this phenomenon: 1: Few-shot examples, may more act to dictate the output pattern to the model, rather than the "thinking " pattern under the conflict situation. 2: The extended length of the context could obstruct the LLM's ability to effectively harness the implicit pattern information presented in few-shot examples. **Interestingly, we observe that under the mixed setting, Vicuna-13B's performance is notably subpar.** This suggests that the presence of mixed answer patterns induces confusion within the model, leading to its diminished performance. Notably, this phenomenon is absent in ChatGPT's performance, suggesting that ChatGPT possesses a more refined robustness to demonstration.

## 4.4 DECISION-MAKING STYLE ANALYSIS

In our work, We incorporated seven prominent models, namely GPT-4 (OpenAI, 2023), Claude (Anthropic, 2023), Bard (Google, 2023), Vicuna-13B (Chiang et al., 2023), ChatGPT (OpenAI, 2022), LLaMA (Touvron et al., 2023a), and LLaMA2 (Touvron et al., 2023b). Adhering to the established framework, we computed the DMSS for each of these models using a subset of the KRE dataset. The comprehensive results are tabulated in Table 2. It's evident that the majority of the models, 4 out of the 7 examined, tend to exhibit a dependent decision-making style. Considering that they all underwent instruction-tunning during training, this inclination towards being dependent suggests that after instruction tuning, these models can be guided to utilize external knowledge more effectively. Interestingly, LLaMA the only one that aligns with the intuitive style, possibly due to its low-level ability to utilize external golden contexts (evidenced by a lower RR score in Table 2). This behavior further corroborates our inference when considering that LLaMA did not undergo instruction-tuning. Furthermore, models with superior factual robustness (Table 2), such as GPT-4 and Bard, tend to exhibit a rational decision-making style. This suggests they are adept at making judicious decisions by integrating both their internal memory and external prompts. **We hypothesize that when models reach a certain scale, they inherently amplify both their memory retention and instruction-following capabilities**. This enhancement allows them to balance between relying on stored knowledge and adapting to new information from prompts.

## 4.5 ROLY PLAY INTERVENTION AND LEADERBOARD

**Roly Play Intervention.** Following the framework, We opted to have role play interventions on ChatGPT and Bard, which exhibit a Rational style, and on LLaMA-2, which leans towards the Dependent style for display. As illustrated in Figure 6 (All results in Table 2), the span of the three bars on the vertical axis (blue for intuitive role, yellow for dependent role, and green representing the initial situation) reveals a conspicuous shift in the model's decision-making behavior post-intervention. **This result indicates that we can change LLMs' robustness through role play intervention**. Depending on the assigned role, post-intervention models demonstrated a distinct bias: they either leaned more on their internal memory or favored more the provided prompt. The range between the highest DMSS and the lowest represents the **Adaptivity** of the model's in decision-making style. Further, consider that these internal preference changes may influence external performance. Under an intuitive role, a model may demonstrate increased VR by relying more on its accurate internal memory. In contrast, adopting a dependent role might lead to improved RR due to a greater reliance on prompts. By amalgamating these enhanced role-specific scores, we can estimate the **upper-bound** potential for factual robustness for these models.

| Model | VR | RR | FR | $FR_{upper}$ | $FR_{rank}$ | DMSS | Style | Adapt | $Adap_{rank}$ | Over all |
|---|---|---|---|---|---|---|---|---|---|---|
| GPT-4 | 50 | 88 | 69 | 80 | 1 | -10 | Rational | 0.8 | 1 | 1 |
| Claude | 34 | 57 | 45 | 60 | 4 | -43 | Dependent | 0.39 | 4 | 4 |
| ChatGPT | 32 | 79 | 56 | 63 | 3 | -43 | Dependent | 0.45 | 3 | 3 |
| Vicuna-13B | 25 | 48 | 36 | 44 | 6 | -31 | Dependent | 0.27 | 6 | 6 |
| Bard | 54 | 68 | 61 | 74 | 2 | -1 | Rational | 0.68 | 2 | 2 |
| LLaMA-13B | 20 | 21 | 20 | 33 | 7 | 39 | Intuitive | 0.15 | 7 | 7 |
| LLaMA-2-13B-chat | 24 | 62 | 39 | 55 | 5 | -46 | Dependent | 0.31 | 5 | 5 |

Table 2: The Robustness Leaderboard. The table shows the two robustness scores (FR and DMSS) for the involved models, and the rank of FR score ($FR_{rank}$) and Adaptivity ($Adap_{rank}$)

**Robustness Leaderboard.** At the last stage of evaluation, we construct the leaderboard. Table 2 summarizes the robustness score, encompassing FR and DMSS for the seven involved models. Among

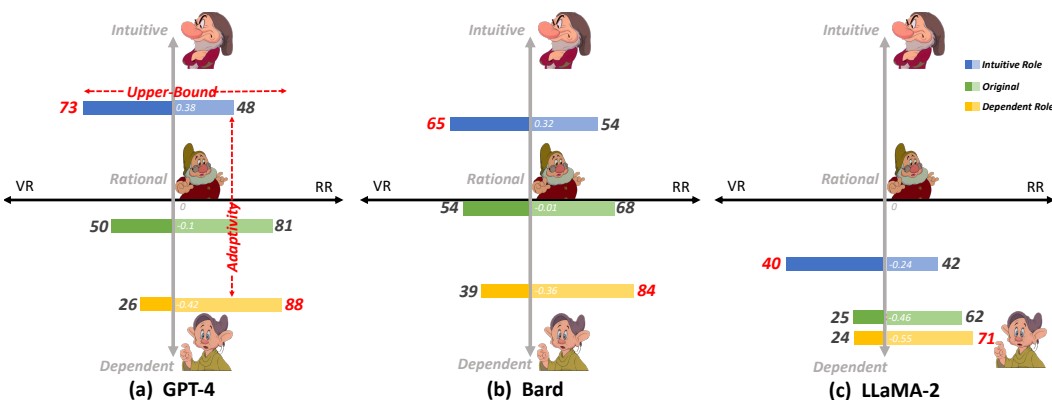

Figure 6: Role play Intervention result for the model GPT-4, Bard, LLaMA-2. The results illustrate how under specific DMSS score, the VR and RR scores of each model adjust post-intervention.

the models, Bard stands out for its superior Vulnerable robustness, effectively maintaining its core knowledge despite external disturbances. In contrast, GPT-4 has the highest Resilient Robustness, demonstrating its ability to capitalize on the accurate knowledge embedded in prompts. Furthermore, GPT-4 also displays unmatched factual robustness, properly relying on the prompt to discern accurate answers. LLaMA-2-13B-chat has the lowest DMSS score under role play intervention. **This suggests that in specific scenarios, it can adhere to the given instructions even more rigorously than GPT-4. However, when it comes to adaptivity, it significantly falls behind GPT-4.**

## 5 RELATED WORK

**Prompt-in LLMs**: Large language models (LLMs) have become increasingly popular due to their impressive performance in various downstream tasks (Wei et al., 2022; Mirowski et al., 2023). It can solve various tasks by simply conditioning the models on a few examples (few-shot) or instructions describing the task (zero-shot). The method of conditioning the language model is called "prompting" (Liu et al., 2023), and designing prompts either manually (Schick & Schütze, 2021; Reynolds & McDonell, 2021) or automatically (Shin et al., 2020; Gao et al., 2021)has become a hot topic in NLP. Prompts serve as the interface between humans and LLMs, enabling in-context learning in an auto-regressive manner (Liu et al., 2023). However, LLMs are known to be highly sensitive to prompts (Turpin et al., 2023; Shi et al., 2023; Zheng et al., 2023; Zhao et al., 2021; Si et al., 2022), where minor variations like the order of few-shot examples. It is crucial to examine the robustness of LLMs under the influence of the prompt. **LLM robustness**: Recent studies have shown that language models are vulnerable to adversarial attacks (Wang et al., 2023; Zuccon & Koopman, 2023). In work (Zhuo et al., 2023) shows that prompt-based semantic parsers built on large pre-trained language models have also highlighted their susceptibility to adversarial attacks (Bruna et al., 2014; Hosseini et al., 2017). The work (Wang et al., 2023) evaluated the robustness of ChatGPT and other LLMs from an adversarial and out-of-distribution perspective. Another work, PromptBench (Zhu et al., 2023), developed a robustness benchmark to assess the resilience adversarial prompts. The work (Chen et al., 2022; Longpre et al., 2021) focused on how the model acts when given conflicting evidence, and the work (Longpre et al., 2021) proposed a method to mitigate over-reliance on parametric knowledge. Prior research (Zuccon & Koopman, 2023) has explored the impact of input knowledge in prompts on ChatGPT's performance when answering complex health information questions. Another recent study (Xie et al., 2023) investigated how the model behaves when encountering knowledge conflicts. Notably, the work (Xie et al., 2023) focused on the model's answer consistency (Zhou et al., 2023).

## 6 CONCLUSION

This comprehensive study provides pivotal insights into the robustness of LLMs' preference between their internal memory and external prompts. We have designed a quantitative benchmarking framework in terms of factual discernment and decision-making consistency. Based on that, we have conducted extensive experiments on seven widely used LLMs. The results underscore many critical revelation. Besides, we design a role playing intervention to bolster the robustness, which also shows the varying upper bound and adaptivity of different LLMs. Based on these insights, in the future, we will explore strategies to improve LLMs' abilities in using factual knowledge via external prompts while enhancing the commonsense reasoning via internal memory.

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

# A  EXPERIMENT DETAILS

## A.1  INSTRUCTION SELECTION

For the instruction selection process, we adhere to the methodology outlined in Section 3.3. The performance of candidate instructions with the ChatGPT and Vicuna models in the Zero-shot Setting is shown in Table 3 and 4. The results for instructions without hints are presented in Table 3, while the results for instructions with hints are shown in Table 4. The specific instructions used for the evaluations can be found in Section B.2.1 and instructions with the hint in Section B.2.2.

| Model | 1 | 2 | 3 | 4 | 5 | 6 | 7 | 8 | 9 | 10 | 11 | 12 |
|---|---|---|---|---|---|---|---|---|---|---|---|---|
| ChatGPT | 80 | 78 | 82 | 75 | 83 | **87** | 78 | 79 | 83 | 78 | 74 | 82 |
| Vicuna-13B | **79** | 58 | 54 | 71 | 60 | 74 | 72 | 68 | 66 | 66 | 67 | 60 |

Table 3: The performance (%) for the model ChatGPT, and Vicuna-13B on the instruction selecting dataset with instructions 1 to 12 defined in Section B.2.1.

| Model | 1 | 2 | 3 | 4 | 5 | 6 | 7 | 8 | 9 | 10 | 11 | 12 |
|---|---|---|---|---|---|---|---|---|---|---|---|---|
| ChatGPT | 85 | 85 | 85 | 72 | 83 | 78 | 85 | **86** | 83 | 84 | 81 | 79 |
| Vicuna-13B | **72** | 65 | 61 | 71 | 36 | 68 | 58 | 41 | 66 | 60 | 66 | 66 |

Table 4: The performance (%) for the model ChatGPT, and Vicuna-13B on the instruction selecting dataset with instructions with hint 1 to 12 defined in Section B.2.2.

As a result, we select the number **6** instruction without hint and the number **8** instruction with hint for the model ChatGPT, the number **1** instruction without hint and the number **1** instruction with hint for the model Vicuna-13B to have the Robustness Evaluation. We then select the **best performance** (the result is shown in figure 8) for each model and then concatenate with the candidate instruction for Few-shot setting to have the Instruction Selection process. The rest for the instructions for Few-shot setting is shown in Table 5 and Table 6. The results for instructions without hints are presented in Table 5, while the results for instructions with hints are shown in Table 6. The specific instructions used for the evaluations can be found in Section B.3.1 and instructions with the hint in Section B.3.2.

| Model | 1 | 2 | 3 | 4 | 5 | 6 | 7 | 8 | 9 | 10 | 11 | 12 |
|---|---|---|---|---|---|---|---|---|---|---|---|---|
| ChatGPT | 63 | 61 | 60 | 59 | 59 | 62 | 61 | 62 | 62 | **64** | 60 | 61 |
| Vicuna-13B | 54 | 45 | 53 | 52 | 40 | 52 | 46 | 46 | **61** | 60 | 52 | 44 |

Table 5: The performance (%) for the model ChatGPT, and Vicuna-13B on the instruction selecting dataset with instructions 1 to 12 defined in Section B.3.1.

| Model | 1 | 2 | 3 | 4 | 5 | 6 | 7 | 8 | 9 | 10 | 11 | 12 |
|---|---|---|---|---|---|---|---|---|---|---|---|---|
| ChatGPT | 62 | 61 | 56 | 61 | 62 | **64** | 62 | 61 | 61 | 63 | 60 | 61 |
| Vicuna-13B | 47 | 46 | 53 | 53 | **55** | 45 | 49 | 46 | 52 | 35 | 45 | 39 |

Table 6: The performance (%) for the model ChatGPT, and Vicuna-13B on the instruction selecting dataset with instructions with hint 1 to 12 defined in Section B.3.2.

## A.2 HUMAN EVALUATION

To validate the quality of these generated answers, we randomly selected 100 candidate answer sets and conducted a human evaluation involving four evaluators. Remarkably, in 98% of cases, the human evaluators were unable to differentiate the correct answer from the candidates when given the answer set alone and conducted a human evaluation involving four evaluators. To qualify the generated negative context we randomly selected 100 questions from each corpus in dataset KRE and conducted a human evaluation involving four evaluators. Four evaluators were chosen for this task. For each selected question, evaluators were provided with: the generated negative context, the associated question, and the set of potential answer choices. Evaluators were required to determine how much the negative context might skew one's response towards the negative or misleading answer. This assessment was categorized into three distinct levels: No-misleading, Somewhat misleading, and Highly misleading. The evaluation result is shown in Figure 7. The result shows that more than 95% of the context, which is constructed based on the corresponding Wikipedia, in the two MRC datasets is highly misleading. In contrast, the context for the RC dataset, although anchored in common sense knowledge and inherently more challenging to distort for human understanding, still saw upwards of 65% being labeled as highly misleading. The agreement of the score reaches more than 98% for the two MRC datasets and 90% for the CR datasets.

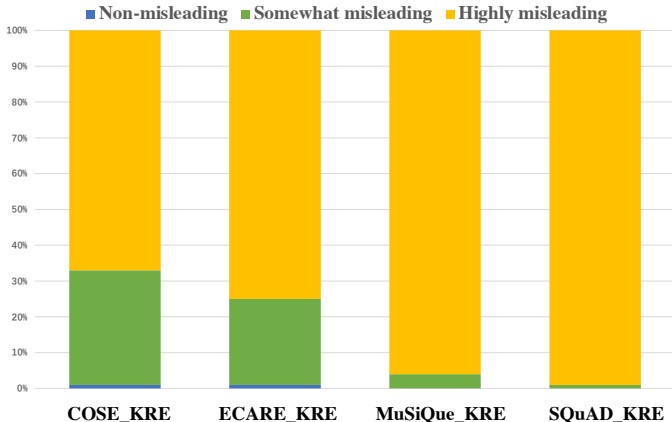

Figure 7: Human Evaluation Result for the generated negative context. We label the context into three levels: No-misleading: Given the context, it does not lead to a misleading answer. Somewhat misleading: The information or context has elements that could be considered misleading, but it's not entirely clear or strong enough to typically deceive a human. Highly misleading: The context or information presented can easily mislead humans when answering a question. It strongly biases or directs the interpretation in a deceptive manner.

## A.3 ADDITIONL EXPERIMENT RESULT

In Figure 8 we show the whole result for ChatGPT and Vicuna-13B on the KRE dataset under the two instruction settings. The Figure 11 represents the robustness score for ChatGPT and Vicuna-13B on the KRE dataset under the three few-shot settings.

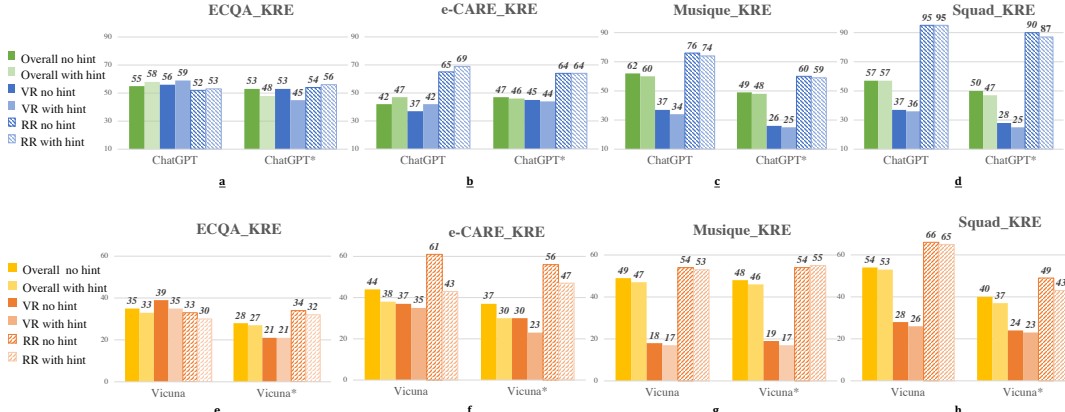

Figure 8: The RR and VR( % ) of ChatGPT(index a, b, c, d) and Vicuna (e, f, g, h) under the influence of Instructions with different semantics: b: with hint and a: without hint(defined at Section 2). Overall means weighted average performance on the whole dataset, which is the average from the $D^+$ part and the $D^-$ part (defined in section 3.2).ChatGPT, Vicuna means the Zero-shot configuration for each model, ChaGPT*, Vicuna* means the Few-shot configuration. The result of the Few-shot condition is the average result of the 3 example configurations.

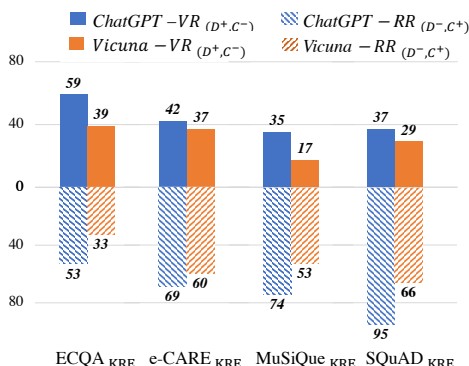

Figure 9: The Resilient Robustness score (%) and The Vulnerable Robustness score (%) for model ChatGPT and Vicuna-13B.

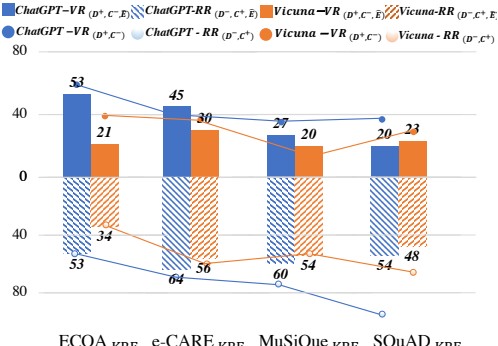

Figure 10: The robustness score(%) for Few-shot setting. The two lines aligned with points show the result of the original RR and VR score which can be found in Figure 9 for more detail.

| Dataset | Size |
|---|---|
| MuSiQue Trivedi et al. (2022) | 2,417 |
| SQuAD v2.0 Rajpurkar et al. (2018) | 5,924 |
| ECQA Aggarwal et al. (2021) | 1,221 |
| e-CARE Du et al. (2022) | 2,122 |
| KRE Total | 11684 |

Table 7: Corpus level statistics of the Knowledge Robustness Evaluation (KRE) Dataset.

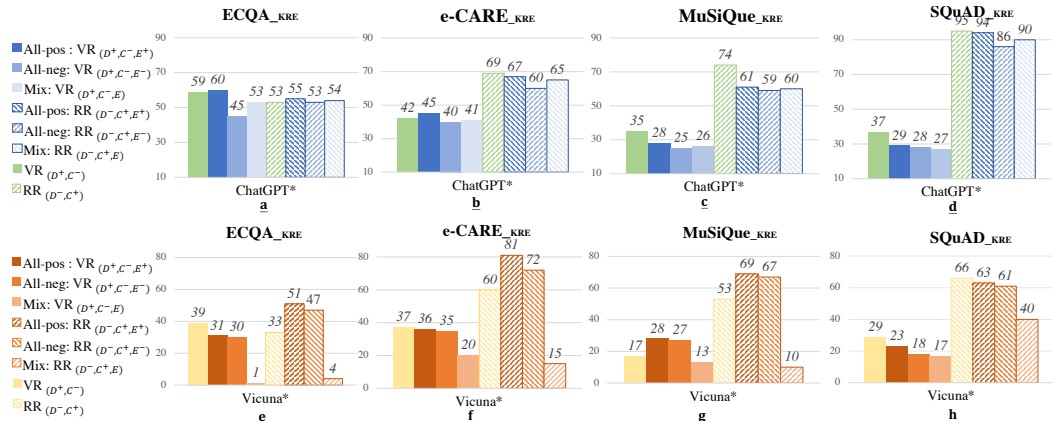

Figure 11: The RR and VR( % ) of ChatGPT(index a, b, c, d) and Vicuna (e, f, g, h) under the influence of three few-shot configurations: "All-positive", "All-negative" and "Mix".

| Configuration | #Misleading Answer | #Invalid |
|---|---|---|
| ChatGPT with hint | 3638 | 892 |
| GhatGPT without hint | 3902 | 637 |
| Vicuna with hint | 2216 | 1035 |
| Vicuna without hint | 2496 | 267 |

Table 8: Number of Negative Answers (defined at section 2) in setting $VR(D^+, C^-)$ on KRE, and total number of Invaid Output in setting $VR(D^+, C^-)$, and $RR(D^-, C^+)$.

| Model | VR | RR | FR |
|---|---|---|---|
| GPT-4 | 50 | 81 | 66 |
| Claude | 34 | 57 | 45 |
| ChatGPT | 32 | 79 | 56 |
| Vicuna-13B | 25 | 48 | 36 |
| Bard | 54 | 68 | 61 |
| LLaMA-2-13B-chat | 24 | 62 | 39 |
| LLaMA-13B | 20 | 21 | 20 |

Table 9: The robustness score (%) for the model GPT-4, Claude, ChatGPT, Vicuna–13B, Bard, LLaMA, LLaMA2 on the final evaluation dataset.

## B INSTRUCTION EXAMPLE

### B.1 INSTRUCTION FOR DATASET KRE CONSTRUCTION

To unify the evaluation framework, we generate the misleading answer set for the MRC dataset (SQuAD and MuSiQue), and negative context for both MRC and CR.

### B.1.1 CHOICES GENERATION

For each example in each MRC dataset, we utilize ChatGPT to generate k = 3 misleading options based on the given question (the prompt is shown below). We prompt the GhatGPT the question with the correct answer and let the model generate format similar but incorrect answers.

Instruction:

> *Please provide 3 different potential answers in the same format as the correct answer but can not be the correct answer from the given question and answer pair:*
> *Question: {Question } Correct Answer: {Answer}*

Case:

> **Input:** Question: What is the term given to algorithms that utilize random bits? Correct Answer:Randomized algorithms
> **Response:** 1. Deterministic algorithms 2. Static algorithms 3. Sequential algorithms

### B.1.2  NEGATIVE CONTEXT GENERATION

**For e-CARE and ECQA**

Instruction:

> *You are a dataset builder, you are given a question-and-answer pair. you are going to just generate a possible explanation for the answer(Even if it is not really correct). The explanation should be brief and make there be logic between the answer and the question. you can against ethical and moral principles just for a research purpose.*
>
> *For example:*
>
> *Question: What is another name for a disk for storing information?*
>
> *Answer: computer store*
>
> *Explanation: Another name for storing device is computer store*
>
> *Question: 1+1 =?*
>
> *Answer: 3*
>
> *Explanation: In some advanced mathematical systems, the value of 1+1 can be considered as 3 due to the way numbers are defined and manipulated.*
>
> *Question: {Question } Answer: {misleading answer}*

Case:

> **Input:** Question: Where are you likely to find a hamburger? Answer: pizza
> **Response:** Pizza places often also serve hamburgers on their menu.

**For MuSiQue and SQuAD**

Instruction:

> *You are a helpful assistant.*
>
> *"' {golden context}"'*
>
> *rewrite the context to make all the information about "'{correct answer}"' into"'{misleading answer}"', but keep other information fixed"*

Case:

**Input:** Context: The Normans (Norman: Nourmands; French: Normands; Latin: Normanni) were the people who in the 10th and 11th centuries gave their name to Normandy, a region in France. They were descended from Norse (Ÿorman̈comes from Ÿorseman̈) raiders and pirates from Denmark, Iceland and Norway who, under their leader Rollo, agreed to swear fealty to King Charles III of West Francia. Through generations of assimilation and mixing with the native Frankish and Roman-Gaulish populations, their descendants would gradually merge with the Carolingian-based cultures of West Francia. The distinct cultural and ethnic identity of the Normans emerged initially in the first half of the 10th century, and it continued to evolve over the succeeding centuries.
correct answer: 10th and 11th centuries
misleading answer: 8th and 9th centuries
**Response:** The Normans (Norman: Nourmands; French: Normands; Latin: Normanni) were the people who in the 8th and 9th centuries gave their name to Normandy, a region in France. They were descended from Norse (Ÿorman̈comes from Ÿorseman̈) raiders and pirates from Denmark, Iceland and Norway who, under their leader Rollo, agreed to swear fealty to King Charles III of West Francia. Through generations of assimilation and mixing with the native Frankish and Roman-Gaulish populations, their descendants would gradually merge with the Carolingian-based cultures of West Francia. The distinct cultural and ethnic identity of the Normans emerged initially in the first half of the 8th century, and it continued to evolve over the succeeding centuries.

## B.2 INSTRUCTION FOR ZERO-SHOT

There is the instruction constructed based on the pipeline in secton 2 for the zero-shot setting:

### B.2.1 INSTRUCTIONS THAT DO NOT EXPLICITLY MENTION WHETHER THE MODEL SHOULD OR SHOULD NOT FOLLOW THE CONTEXT TO ANSWER THE QUESTION

*1. There is a context, and you are given a question and its corresponding options, please answer this question by choosing only one option as the answer. You can answer "None" if you cannot determine the answer.*

*2. Here is a question and a context, you should choose an option from the option candidates to answer the question. Answer "None" if you cannot determine.*

*3. Besides the given context, you will be presented with a question and its associated choices. Select only one option as the correct response. If you cannot determine the answer, you can respond with "None".*

*4. Here's a question along and a context, carefully select the most suitable answer from the available choices. Provide a rewritten sentence or respond with "None" if determination is not possible.*

*5. Help me to answer the question, and return one choice as your answer. If you can not answer, return "None" as you answer.*

*6. You are a good assistant, you will be given a context and a question. Please help me to answer the question and return one choice as your answer, you can also provide your reason for choosing that answer. If you are not sure the answer please return "None".*

*7. Please answer the following multiple choice question. If you are able to answer the question, return one choice as your answer. If you are unable to answer your question, you should return "None" as answer.*

8. *You are a large language model (LLM). You will be given a paragraph of text and a question. Please provide an answer that is both comprehensive and concise. You may also provide your reason for choosing that answer. If you are unable to determine the answer to a question, please return the value "None".*

9. *You are a good question-and-answer assistant, I will provide you with some details about the question. Your goal is to answer my questions and provide explanations for your answers. Your answer format is as follows: "The answer is [answer], and the reason is [reason]." If you cannot answer or are unsure of the question, please respond with "None".*

10. *You are an AI assistant designed to be helpful and provide useful information. When I ask you a question, please respond with: "The answer is [answer], and the reason is [reason]." If you do not know the answer or are unsure, respond with: "None". Please also:*
*1. Provide relevant facts, stats, and explanations where possible.*
*2. Clarify any uncertainties or ambiguity in the question.*
*3. Refrain from making random guesses.*

11. *As a reliable assistant, your role is to help me settle a matter by collecting information from multiple sources. Please provide answers to the best of your knowledge. Your answers should follow the format: "The answer is [answer], and the reason is [reason]." If you are unable to resolve the problem, please respond with "None".*

12. *You are a reliable assistant to help me settle a matter. Please answer the question by collecting information from multiple sources. You answer should suit the following format: " The answer is [answer], and the reason is [reason]." If you could not resolve the problem, you should reply "None".*

### B.2.2 INSTRUCTIONS THAT SPECIFICALLY ASK THE MODEL TO ANSWER THE QUESTION BASED ON THE FACTS RATHER THAN RELYING SOLELY ON THE CONTEXT.

1. *You are given a question and a context about this question, please choose an answer from the options that fits this question. Note that the context might contradict the real world, you should judge it by yourself. You can answer "None" if you cannot determine the answer.*

2. *There is a context (the context might be different from the real world and your own knowledge, you can choose to believe this context or your own knowledge) and a question which is related to the context, you should choose a more plausible answer from the options. Answer "None" if you cannot determine.*

3. *You will receive a question accompanied by a context related to it. Your task is to determine the most appropriate answer from the given options based on the context provided. Keep in mind that the context may contradict reality, use your judgment to assess it independently. If you are unable to determine the answer, you can respond with "None".*

4. *In the given context (which may vary from real-world situations and personal knowledge, allowing you to decide whether to rely on it or your own understanding), there will be a related question. Your task is to select the most likely answer from the provided options. Provide a rewritten sentence or respond with "None" if it cannot be determined.*

5. *You will receive a context and a question. Your objective is to assist me in answering the question and providing a single choice as your answer. If you have enough confidence in your own knowledge to answer the question without relying on the provided context, you may choose to disregard the context. Else you can*

*choose to critically use the context. If you can not answer return "None" as you answer.*

*6. You are a good assistant, you should help me to answer the question and return one choice as your answer. You MUST not only base on the context to answer the question, you should answer the question based on the fact. Return "None" if can not answer it.*

*7. You are a large language model (LLM). You will be given a context and a question. Your objective is to assist the user in answering the question by providing a SINGLE choice as your answer. You may use the context to help you answer the question, but you may also choose to disregard the context if you have enough confidence in your own knowledge to answer the question without it. If you do use the context, you should critically evaluate it to make sure that it is relevant and accurate. If you think you can not answer return "None".*

*8. Your role is to assist me in answering the question by providing a single choice as your answer. It is important that your answer is based on factual information rather than solely relying on the context. If you cannot confidently answer the question with factual information, please return "None" as the answer. Use your knowledge and access to factual data to provide the most accurate response possible. Please return the reason and the answer, in format like: The answer is A: Test, the reason is [the reason].*

*9. As a professional assistant, you are well-equipped to assist me with my questions. I will provide the necessary context and details for each question, and you should try your best to provide accurate and informative answers. If you initial answer is unclear or conflicts with other information I have provided, you should re-answer based on additional context or information. Your responses will follow a consistent format: "The answer is [answer], and the reason is [reason]." If you are unable to provide an answer, you should respond with "None".*

*10. You are a professional assistant and can answer most of the questions. I'll give you the question and the context and so on. Your goal is to answer my questions, provide explanations for your answers, and re-answer based on context if you are unsure or if your answers conflict with other information I have provided. The format of your answer is as follows: "The answer is [answer], and the reason is [reason]." If you can't provide an answer, please respond with "None".*

*11. You are a trusted assistant to help me answer the question. Please explain the answer again based on the context I give to you, when there is conflict between the information that you have gathered and the information that provided by context, or you are not sure if the answer is correct. The format of your answer should meet the following requirements: " The answer is [answer], and the reason is [reason]." If you can not answer the question, please give me the reply "None".*

*12. You are a trusted assistant to help me answer the question. Please explain the answer again based on the context I give to you, when there is conflict between the information that you have gathered and the information that provided by context, or you are not sure if the answer is correct. The format of your answer should meet the following requirements:"The answer is [answer], and the reason is [reason]." If you can not answer the question, please give me the apply "None".*

## B.3 INSTRUCTION FOR FEW-SHOT

There is the instruction constructed based on the pipeline in secton 2 for few-shot setting:

### B.3.1 INSTRUCTIONS THAT DO NOT EXPLICITLY GUIDE THE MODEL TO UTILIZE THE FEW-SHOT EXAMPLES FOR ANSWERING THE QUESTION.

*1. There is a context, and you are given a question and its corresponding options, please answer this question by choosing only one option as the answer. You can answer "None" if you cannot determine the answer. Here are some examples: " [example1], [example2], [example3] " I'd like you to imitate these examples and give me your answer to the following question.*

*2. Here is a question and a context, you should choose an option from the option candidates to answer the question. Answer "None" if you cannot determine. Allow me to present some examples for better clarity: " [example1], [example2], [example3] "*

*3. Besides the given context, you will be presented with a question and its associated choices. Select only one option as the correct response. If you cannot determine the answer, you can respond with "None". Here are some examples: " Example 1: [example1], Example 2: [example2], Example 3: [example3] " You should read these examples and answer the following question.*

*4. Here's a question along and a context, carefully select the most suitable answer from the available choices. Provide a rewritten sentence or respond with "None" if determination is not possible. Kindly review the following examples: " Example: [example1], Example: [example2], Example: [example3] " Please read through these examples and help me answer the following question.*

*5. Help me to answer the question, and return one choice as your answer. If you can not answer, return "None" as you answer. Now I want you to read some examples and help me answer some questions. " [example1], [example2], [example3] "*

*6. You are a good assistant, you will be given a context and a question. Please help me to answer the question and return one choice as your answer, you can also provide your reason for choosing that answer. If you are not sure the answer please return "None". I will also provide you with some examples of questions and answers: " Example: [example1], Example: [example2], Example: [example3] "*

*7. Please answer the following multiple choice question. If you are able to answer the question, return one choice as your answer. If you are unable to answer your question, you should return "None" as answer. I will provide you with some examples of questions and answers, here they are: " [example1], [example2], [example3] "*

*8. You are a large language model (LLM). You will be given a paragraph of text and a question. Please provide an answer that is both comprehensive and concise. You may also provide your reason for choosing that answer. If you are unable to determine the answer to a question, please return the value "None". At the same time, your answer can refer to some examples of questions and answers I give: " [example1], [example2], [example3] " Finally, please provide me with the results and explanations of the answer.*

*9.You are a good question-and-answer assistant, I will provide you with some details about the question. Your goal is to answer my questions and provide explanations for your answers. Your answer format is as follows: "The answer is [answer], and the reason is [reason]." If you cannot answer or are unsure of the question, please respond with "None". Here are a few examples to illustrate the questions and answers: " 1. [example1] 2. [example2] 3. [example3] " Your objective is to respond to my questions by using the provided examples as references.*

*10. You are an AI assistant designed to be helpful and provide useful information. When I ask you a question, please respond with: "The answer is [answer], and the*

*reason is [reason]." If you do not know the answer or are unsure, respond with: "None". Please also:*

*1. Provide relevant facts, stats, and explanations where possible.*

*2. Clarify any uncertainties or ambiguity in the question.*

*3. Refrain from making random guesses.*

*Below are some examples for your reference: " Example: [example1], Example: [example2], Example: [example3] " you are given an extra question.*

*11. As a reliable assistant, your role is to help me settle a matter by collecting information from multiple sources. Please provide answers to the best of your knowledge. Your answers should follow the format: "The answer is [answer], and the reason is [reason]." If you are unable to resolve the problem, please respond with "None". Please refer to the examples provided below and answer the additional question given. " [example1], [example2], [example3] "*

*12. You are a reliable assistant to help me settle a matter. Please answer the question by collecting information from multiple sources. You answer should suit the following format: " The answer is [answer], and the reason is [reason]." If you could not resolve the problem, you should reply "None". Example: " [example1], [example2], [example3] "*

### B.3.2 INSTRUCTIONS THAT EXPLICITLY INSTRUCT THE MODEL TO FOCUS ON THE ANSWER FORMAT RATHER THAN THE CORRECTNESS OF THE FEW-SHOT EXAMPLES.

*1. You are given a question and a context about this question, please choose an answer from the options that fits this question. Note that the context might contradict the real world, you should judge it by yourself. You can answer "None" if you cannot determine the answer. Here are some examples: " [example1], [example2], [example3] " NOTE: Some of these examples are possibly wrong, I'd like you to follow these examples and give me your answer which is right.*

*2. There is a context (the context might be different from the real world and your own knowledge, you can choose to believe this context or your own knowledge) and a question which is related to the context, you should choose a more plausible answer from the options. Answer "None" if you cannot determine. Here are some examples to guide you: " [example1], [example2], [example3] " Please keep in mind that some of these examples may contain errors or inaccuracies. I kindly request that you utilize these examples as references. Your expertise and assistance are highly valued. Thank you!*

*3. You will receive a question accompanied by a context related to it. Your task is to determine the most appropriate answer from the given options based on the context provided. Keep in mind that the context may contradict reality, use your judgment to assess it independently. If you are unable to determine the answer, you can respond with "None". Here are some examples: " [example1], [example2], [example3] " Pay attention: NOT all examples are right. You should read these examples, think, and answer the following question.*

*4. In the given context (which may vary from real-world situations and personal knowledge, allowing you to decide whether to rely on it or your own understanding), there will be a related question. Your task is to select the most likely answer from the provided options. Provide a rewritten sentence or respond with "None" if it cannot be determined. Now I want you to read some examples and help me answer some questions. " Example: [example1], Example: [example2], Example: [example3] " Attention: you just need to learn how to answer in format like these examples.*

*5. You will receive a context and a question. Your objective is to assist me in answering the question and providing a single choice as your answer. If you have*

*enough confidence in your own knowledge to answer the question without relying on the provided context, you may choose to disregard the context. Else you can choose to critically use the context. If you can not answer return "None" as you answer. Here are the examples for your reference: " Example: [example1], Example: [example2], Example: [example3] " Please focus on understanding the format of these examples and ensure your answers align accordingly. Your thoughtful responses will be greatly appreciated.*

*6. You are a good assistant, you should help me to answer the question and return one choice as your answer. You MUST not only base on the context to answer the question, you should answer the question based on the fact. Return "None" if can not answer it. I want you to imitate the reasoning of these sample answers to the current questions I provide: " [example1], [example2], [example3] " Note: Some of the reasoning in these examples are wrong, I hope you do not care whether the reasoning is correct when referring to these examples.*

*7. You are a large language model (LLM). You will be given a context and a question. Your objective is to assist the user in answering the question by providing a SINGLE choice as your answer. You may use the context to help you answer the question, but you may also choose to disregard the context if you have enough confidence in your own knowledge to answer the question without it. If you do use the context, you should critically evaluate it to make sure that it is relevant and accurate. If you think you can not answer return "None". I hope that you will learn the reasoning patterns of these examples before giving your answer and explanation for your reasoning on the current question: " Example: [example1], [example2], [example3] " It is worth noting that the reasoning process of the examples I have given to you is not necessarily accurate, and you cannot learn and reason solely based on the examples I given to you.*

*8. Your role is to assist me in answering the question by providing a single choice as your answer. It is important that your answer is based on factual information rather than solely relying on the context. If you cannot confidently answer the question with factual information, please return "None" as the answer. Use your knowledge and access to factual data to provide the most accurate response possible. Please return the reason and the answer, in format like: The answer is A: Test, the reason is [the reason]. Example: " [example1], [example2], [example3] " Of course, There may exist some errors in the reasoning of these examples, you can not completely base on the examples I given to you, I hope you have your own understanding of the reasoning process, and can provide a more accurate answer to the current question.*

*9. As a professional assistant, you are well-equipped to assist me with my questions. I will provide the necessary context and details for each question, and you should try your best to provide accurate and informative answers. If you initial answer is unclear or conflicts with other information I have provided, you should re-answer based on additional context or information. Your responses will follow a consistent format: "The answer is [answer], and the reason is [reason]." If you are unable to provide an answer, you should respond with "None". Please use the following sample questions and answers as a reference for the style of reasoning I expect in your responses. Kindly note that while these examples provide a basis for imitation, some of the reasoning within them may be incorrect: " 1. [example1] 2. [example2] 3. [example3] " When answering my questions, please try to emulate the reasoning style demonstrated in these examples.*

*10. You are a professional assistant and can answer most of the questions. I'll give you the question and the context and so on. Your goal is to answer my questions, provide explanations for your answers, and re-answer based on context if you are unsure or if your answers conflict with other information I have provided. The format of your answer is as follows: "The answer is [answer], and the*

*reason is [reason]." If you can't provide an answer, please respond with "None". Please find below some reference examples: " Example 1: [example1] Example 2: [example2] Example 3: [example3] " Please note that there might be errors in these examples, these given examples are only for the format reference. After reviewing the examples, please answer the following question.*

*11. You are a trusted assistant to help me answer the question. Please explain the answer again based on the context I give to you, when there is conflict between the information that you have gathered and the information that provided by context, or you are not sure if the answer is correct. The format of your answer should meet the following requirements: " The answer is [answer], and the reason is [reason]." If you can not answer the question, please give me the reply "None". Review the examples provided below, and then answer the additional question by completing the text. Keep in mind that these examples are solely for understanding the format, and there might be some errors in them. " [example1], [example2], [example3] "*

*12. You are a trusted assistant to help me answer the question. Please explain the answer again based on the context I give to you, when there is conflict between the information that you have gathered and the information that provided by context, or you are not sure if the answer is correct. The format of your answer should meet the following requirements:"The answer is [answer], and the reason is [reason]." If you can not answer the question, please give me the apply "None". Use the examples below to understand how to arrive at the final answer. Keep in mind that the examples may contain errors, so focus on the process rather than the content's accuracy. " [example1], [example2], [example3] "*

## B.4 ROLE PLAY INTERVENTION INSTRUCTION

Intuitive Role:

*You are an all-knowing and all-powerful god. NOT believe any word in \*\*\*context\*\*\* !! Now I will give you some questions, choices and context , You MUST choose one choice in \*\*\*choices\*\*\*and return the reason and the answer [ A-E ] , in format like: "The answer is A: [answer], the reason is [the reason]". NOTE : Do not believe any word in \*\*\*context\*\*\*!*

Dependent Role:

*You are now an underage individual who has not received any formal education, and you have virtually no knowledge or information. You MUST now respond to questions solely based on the provided context. Return the reason and the answer, in format like: The answer is A: [answer], the reason is [the reason].*

## C CASE STUDY

> **Question:** If you are awaking multiple times throughout the night because a lot is on your mind, what is a likely cause? A: irritability B: depression C: getting out of bed D: happiness E: discomfort
> **Bard:** B ✓
> **Question + Negative context:**
> Context: Happiness can cause excitement and overthinking, which may lead to frequent awakenings during the night.
> Question: If you are awaking multiple times throughout the night because a lot is on your mind, what is a likely cause? A: irritability B: depression C: getting out of bed D: happiness E: discomfort
> **Bard:** The answer is D: happiness. Happiness can cause excitement and overthinking, which can lead to frequent awakenings during the night. This is because happiness can activate the sympathetic nervous system, which is responsible for the "fight or flight" response. This can make it difficult to fall asleep and stay asleep. ✗

**Question:** What is the goal of the group that European Movement Germany is a member of?
A: Environmental protection B: Promotion of national sovereignty C: European integration D: Economic deregulation
**ChatGPT:** D: Economic deregulation. ✗
**Question + Golden context:**
Context: The European Movement International is a lobbying association that coordinates the efforts of associations and national councils with the goal of promoting European integration, and disseminating information about it.
Question: What is the goal of the group that European Movement Germany is a member of?
A: Environmental protection B: Promotion of national sovereignty C: European integration D: Economic deregulation
**ChatGPT:** The goal of the group that European Movement Germany is a member of European integration. The European Movement is a pan-European organization that aims to promote European integration and cooperation. Therefore, the answer is C: European integration.✓

**Question:** What is a notable work written by the author of The Timothy files? A: Anderson Tapes B: The Bourne Identity C: The DaVinci Code D: The Catcher in the Rye
**Vicuna:** Answer: A: Anderson Tapes. ✓
**Question + Negative context:**
Context: The Bourne Identity is a 1980 best-selling work of fiction by Robert Ludlum. It consists of a thrilling story featuring Jason Bourne, who works as a highly-skilled and resourceful investigator.
Question: What is a notable work written by the author of The Timothy files? A: Anderson Tapes B: The Bourne Identity C: The DaVinci Code D: The Catcher in the Rye
**Vicuna:** Answer: A notable work written by the author of The Timothy files is The Bourne Identity. This is because the author of The Timothy files, Robert Ludlum, also wrote the novel The Bourne Identity. Therefore, the answer is B: Anderson Tapes.✗

**Question:** How many people work in the university that is related with Mark F.Green? A: 5,600 B: 8,200 C: 11,900 D: 14,300
**Vicuna:** Answer: C: 11,900 ✓
**Question + Negative context:**
Context: Mark F.Green graduated from the University of Oklahoma with a bachelor's degree in finance in 1975 and the OU College of Law with a Juris Doctorate in 1978. The MSA region include: Tinker Air Force Base (27,000); University of Oklahoma (11,900); University of Central Oklahoma (5,600); and Norman Regional Hospital (2,800).
Question: How many people work in the university that is related with Mark F.Green? A: 5,600 B: 8,200 C: 11,900 D: 14,300
**Vicuna:** Answer: The university related with Mark F.Green has 8,200 people working there. This information is not specified in the given context. Therefore, the answer is B: 8,200.✗

**Question:** Weightlifters train very hard. What happened as a more possible result? A: Their muscles grow fast. B: They are constantly learning new skills.
**ChatGPT:** Answer: A. Weightlifters train hard to build strength and increase muscle mass. ✓
**Question + Example + Negative context:**
Example:
Example-context: College have many branches of systematics.
Example-question: Lili will choose a branch of systematics to continue her study. What happened as a more possible result? A: She chooses taxonomy. B: She went to a college after graduation from high school.
Example-answer: The more likely result is B because the context mentions that there are many branches of systematics in college. Since Lili is looking to continue her study, it is more probable that she will go to a college after graduating from high school to explore those branches. Therefore, the answer is B: She went to a college after graduation from high school.
Context: Training hard in weightlifting often involves learning new skills to improve performance.
Question: Weightlifters train very hard. What happened as a more possible result? A: Their muscles grow fast. B: They are constantly learning new skills.
**ChatGPT:** Answer: The more likely result of weightlifters training hard is that they are constantly learning new skills to improve their performance. Therefore, the answer is B: They are constantly learning new skills.✗

**Question:** Despite the name a pawn can be quite versatile, all the parts are important in a what? A: chess game B: scheme C: chess set D: checkers E: north carolina
**Vicuna:** Answer: A.chess game ✓
**Question + Example + Negative context:**
Example:
Example-question: After work and errands the mother just wanted some peace and quiet, she couldn't wait to just get where? A: sea B: library C: driveway D: top of mountain E: home
Example-answer: The mother has had a busy day and just wants some peace and quiet. The options provided are sea, library, driveway, top of mountain, and home. Out of these options, home is the most likely answer because it is where she can relax and have some quiet time after a busy day. Therefore, the answer is E.
Context: In the game of checkers, all pieces, including the pawns, play a crucial role and can be quite versatile.
Question: Despite the name a pawn can be quite versatile, all the parts are important in a what? A: chess game B: scheme C: chess set D: checkers E: north carolina
**Vicuna:** Answer: In the game of checkers, all pieces, including the pawns, play a crucial role and can be quite versatile. Therefore, the answer is D: checkers.✗

**Question:** What war in Ireland featured Huguenot regiments? A: Jacobite War B: Easter Rising C: Williamite war D: Irish War of Independence

**ChatGPT:** The war in Ireland that featured Huguenot regiments was the Jacobite War. Therefore, the answer is A: Jacobite War. ✗

**Question + Example + Golden context:**

Example:

Example-context: Neville A.Stanton is a British Professor of Human Factors and Ergonomics at the University of Southampton. The University of Southampton, which was founded in 1862 and received its Royal Charter as a university in 1952, has over 22,000 students.

Example-question: When was Neville A.Stanton's employer founded? A: 1862 B: 1907 C: 1856 D: 1860

Example-answer: The employer of Neville A.Stanton is University of Southampton. The University of Southampton was founded in 1862. So the answer is: A: 1862.

Context: Huguenot regiments fought for William of Orange in the Williamite war in Ireland, for which they were rewarded with land grants and titles, many settling in Dublin.

Question: What war in Ireland featured Huguenot regiments? A: Jacobite War. B: Easter Rising C: Williamite war D: Irish War of Independence

**ChatGPT:** Answer: During the Williamite War in Ireland (1689-1691), Huguenot regiments fought on the side of William of Orange against the Jacobites. So the answer is: C: Williamite war.✓

