# OpenReview forum: "Intuitive or Dependent? Investigating LLms’ Robustness to Conflicting Prompts"
_ICLR.cc/2024/Conference — ICLR 2024 Conference Withdrawn Submission_

### Official Review · Reviewer_8SLG · 2023-10-30

**Soundness:** 2 fair
**Presentation:** 2 fair
**Contribution:** 3 good
**Rating:** 5
**Confidence:** 3

**Summary:**

This paper investigates the robustness of large language models (LLMs) to conflicting prompts that may contain contrasting information in real-world applications. The paper proposes a quantitative benchmarking framework and conducts role-playing interventions to control LLMs' preferences. The paper also evaluates seven open-source and closed-source LLMs on a knowledge robustness evaluation (KRE) dataset, revealing their varying performance and behavior.

**Strengths:**

The topic is interesting and novel, and the paper makes several valuable contributions, including:

1. Establishing a comprehensive benchmarking framework that comprises a KRE dataset, a robustness evaluation pipeline, and corresponding metrics.
2. Defining two types of factual robustness (vulnerable and resilient) and three types of decision styles (intuitive, dependent, and rational) to measure LLMs' behavior.
3. Implementing role-play interventions to alter LLMs' robustness and adaptivity by instructing a specific role style.
4. Constructing a robustness leaderboard to compare the performance and capabilities of different LLMs.
5. The experiment results are sufficient enough to support the claims and the presentation of experiment results are clearly.

**Weaknesses:**

However, the paper also has some limitations and weaknesses that could be improved in future work, such as:

1. The instructions for the KRE dataset are generated by 12 hand-crafted prompts written by 4 individuals and then rephrased using ChatGPT. But, it's unclear for me what guidance was provided for generating these instructions. Additionally, the small number of human evaluators (only 4) for human evaluation and of instruction annotators may introduce biases or noise in the data quality, which could affect the robustness evaluation results.
2. The organization of the introduction is challenging to follow, as it introduces several new issues, such as decision style and instructing a role, without clear intuitive connections to the main theme of the paper. Maybe provide some examples to explain these issues would be better.
3. The problem setting appears novel but complex to understand, particularly in terms of motivation. The framework involving negative and positive context, in-context learning, and decision-making (as shown in Figure 1) needs further clarification regarding its motivation. In addition, are a large amount of human-written works required to implement such framework?
4. The evaluation concepts are comprehensive, covering factual robustness and decision-making style scores, but the technical novelty of the evaluation metrics appears to be lacking, primarily relying on frequency-based assessments.

**Questions:**

1. What are role instructions? What the effects of these? And dose role instructions are reasonable and useful in read-world application?
2. How are positive and negative contexts constructed, and what are some application scenarios for factual robustness? Can you provide examples?

Minor issues:
There are some typos, such as "After unraveling the preference."

---

> ### Author Response · Authors · 2023-11-22
> **Response to Reviewer 8SLG (1/2)**
>
> Thank you for your detailed response. Here is the response to your questions
>
> **The guidance and the number of human evaluators setting**
>
> Due to the time and cost limitations, we consider 12 hand-crafted prompts written by 4 individuals.  We try our best to guide the annotators to provide high-quality instructions. In specific, we initially delineated two distinct semantic spaces to guide the instruction creation process. The first set of instructions was designed to elicit model responses without specifying a preference towards the context. The second set contained hints, prompting the model to judiciously evaluate the quality of the context before answering the questions. These instructions are shown in the appendix of our paper.
>
> Regarding the human evaluation aspect, employing four human evaluators is in line with recent practices in LLM evaluation studies [1] [2]. The quality of the constructed dataset is quite high from our human evaluation result shown in the Appendix.
>
> **Decision style, instructing a role, why are they connected to the main theme?**
>
> We make it clear why decision style and role instruction are connected to the main theme. Please refer to the general response.
>
> **Motivation of involving negative and positive context, in-context learning, and decision-making**
>
> 1.Our research primarily focuses on the robustness of Large Language Models in scenarios where there is a conflict between the input context and the model's memory. To effectively construct these conflict scenarios, we deliberately involve both negative and golden/positive contexts (where negative contexts point to the negative answer while the golden/positive context points to the correct answer), which allows us to simulate situations where LLMs might receive conflicting information. (Refer to the general response for more explaintion for the motivation)
>
> 2.In-context learning, in the field of language models like GPT-3, refers to the model's ability to learn and perform tasks based on a few examples provided in its prompt [3]. In-context example is an important part of the prompt, to systematically evaluate robustness we involved this component to guarantee completeness.
>
> 3.The motivations of decision style analysis are: 1). To systemically analyze the preferences models enables us to get insights into the models’ behavioral inclinations. Then we can strategically influence and modify these internal preferences to measure their adaptively and thereby exert control over the model's external performance and get the upper bound of it.
> 2). From a practical perspective, sometimes the conflict between memory and prompts is not related to correctness. That is, both are correct, and it is just about how the models make decisions, e.g., recommendations. Still, this is also important as we want the model's behavior consistent and predictable. 3) Instructing LLMs to play a specific role is commonly used, while there is no quantitative analysis. We are to fill in this research blank. (For more contribution in this aim, please refer to general response).
>
> **The use of frequency-based assessments**
>
> Our contribution is 1. We highlight two kinds of robustness which are all practical in read-world scenarios and 2. Provide a systematic evaluation framework to evaluate the two robustness under conflict. 3.We measure models’ adaptability in role-playing. This aspect of adaptability is particularly crucial, given the growing popularity of role play as a method to direct model behavior, yet it has not been quantitatively assessed before. And also get the upper bound for each model. 4. We have conducted a lot of experiments and got interesting findings.
>
> [1] Zeng, Zhiyuan, et al. "Evaluating large language models at evaluating instruction following." arXiv preprint arXiv:2310.07641 (2023).
>
> [2] Chiang, Cheng-Han, and Hung-yi Lee. "Can Large Language Models Be an Alternative to Human Evaluations?." arXiv preprint arXiv:2305.01937 (2023).
>
> [3] How does in-context learning work? A framework for understanding the differences from traditional supervised learning http://ai.stanford.edu/blog/understanding-incontext/

---

> ### Author Response · Authors · 2023-11-22
> **Response to Reviewer 8SLG (2/2)**
>
> **Role instructions and real-world application**
>
> Role instruction denotes a special instruction in the prompt to change LLMs' behavior to align with human needs, e.g., "As an academic writing assistant,", which is commonly used in real-world applications (As reviewer WVwf mentioned). For instance, users frequently employ prompts such as "You are a helpful assistant" when seeking assistance from ChatGPT. This role instruction is shown as a demonstrative example in OpenAI's documentation[4].
>
> In our work, (Mentioned in Section 3.6) we use two distinct role prompts to steer the models into specific decision-making pathways: Dependent Role: In this intervention, the model is furnished with a role prompt that asks it to prioritize information solely from the external prompt when generating answers.  Intuitive Role: Contrary to the dependent role, this role prompt is designed to push the model towards relying predominantly on its intrinsic memory. (Prompt is shown in Appendix B.4). We want to change the model's inner preference by deploying such instructions. These two prompts are also practical. Taking retrieval augmented LLMs as an example, the retrieval results will be the context and the context may contain the newest information that the model does have. In that case, we expect to use the dependent role instruction to help the model utilize the context.
>
> **Golden and negative  contexts constructed and application scenarios for factual robustness**
>
> The golden/positive contexts are provided in existing datasets, which contain the factual knowledge/commonsense knowledge to answer the question. Negative contexts are modified from positive contexts by corrupting the answer information. Specifically, we employ ChatGPT to replace the answer information in the positive context with the incorrect information for each sample in SQuAD and MuSiQue, For each example in ECQA and e-CARE, we use ChatGPT to generate an explanation for the negative answer as the negative context. The detailed prompt and case are shown in Appendix B.1 (More detailed in Section 2).
>
> Hallucination[5] remains one of the most significant challenges facing Large Language Models. Current research approaches, such as retrieval-augmented methods [6]and tool-augmented techniques[7], aim to mitigate this by prompting models with retrieval results and tool responses (For example, openAI has published plugins to implement on ChatGPT, with which model can help to search on the google to seek for information). However, these retrieval results can introduce conflict (the model lacks/has wrong memory, while the prompt is correct) and sometimes introduce noise or incorrect information. The model's uncertain behavior toward the input prompt may lead to low-quality responses. Our work focuses on models' robustness (factual and decision style) in these conflict situations. By systematically analyzing the robustness of the model we can make this uncertainty clearer thus paving the way to solve hallucination.
>
> [4]demonstrative example for role instruction https://platform.openai.com/docs/guides/text-generation/chat-completions-api
>
> [5] Rawte, Vipula, Amit Sheth, and Amitava Das. "A survey of hallucination in large foundation models." arXiv preprint arXiv:2309.05922 (2023).
>
> [6] Shuster, Kurt, et al. "Retrieval augmentation reduces hallucination in conversation." arXiv preprint arXiv:2104.07567 (2021).
>
> [7]Schick, Timo, et al. "Toolformer: Language models can teach themselves to use tools." arXiv preprint arXiv:2302.04761 (2023).

---

> ### Author Response · Authors · 2023-11-23
> **Response to Reviewer 8SLG**
>
> Dear reviewer:
>
> Thank you for your careful comments again. Since today marks the final day of discussion, we kindly request once again for your feedback and comments. As shown in your comment "I am willing to raise the score for further explanations.", we sincerely look forward to your further feedback. If you find that our responses have effectively addressed your concerns, can you give us feedback? Your feedback is crucial to us. We apologize for any inconvenience caused.

---

### Official Review · Reviewer_WVwf · 2023-11-02

**Soundness:** 3 good
**Presentation:** 2 fair
**Contribution:** 3 good
**Rating:** 5
**Confidence:** 3

**Summary:**

The paper investigates the robustness of Large Language Models (LLMs) when faced with prompts that conflict with their internal memory or present noisy information. The authors introduce a quantitative benchmarking framework to assess this robustness and perform role-playing interventions to determine how LLMs prioritize information. They explore two types of robustness: "factual robustness," which is the model's ability to discern correct facts from prompts or memory, and "decision style," which classifies how LLMs make choices (intuitive, dependent, or rational) in the absence of a clear correct answer.

The study, which tested both open-source and closed-source LLMs, found that these models are prone to being misled by incorrect prompts, particularly regarding commonsense knowledge. The results also suggest that while detailed instructions can reduce the number of misleading answers, they may lead to an increase in invalid responses. By implementing role-play interventions, the authors were able to assess the limits of robustness and adaptability across LLMs of different sizes.

**Strengths:**

- The paper undertakes comprehensive experimental analysis to assess the robustness of Large Language Models (LLMs) when encountering prompts with conflicting information.
- Several conclusions drawn by the paper are perceptive and provide valuable contributions to the community.

**Weaknesses:**

- Certain conclusions presented in the paper appear to necessitate additional discourse and empirical validation.
- The manuscript's writing could substantially benefit from enhancements, as the current draft gives the impression of being hastily prepared.

**Questions:**

- My primary reservation regarding the paper's conclusions pertains to the variability in scores such as DMSS, VR, and RR for LLMs, as depicted in Figure 5. This variability leads to fluctuations in the ranking of models; for instance, Bard's DMSS score falls below GPT-4 in the dependent and original roles but surpasses it in the intuitive role. Similar patterns are observed with VR and RR scores. Given this, one must question whether the paper's conclusions are contingent upon the specific roles and prompts used in the study. Could it be possible that the findings are not universally applicable but rather prompt-specific?

- The use of role-play interventions in the paper does not appear to constitute a novel contribution. This method, which involves instructing LLMs to either consider or disregard context by modifying the prompt, is a common approach in querying LLMs. This leads to a pertinent inquiry: if the intention is for the LLM to overlook the context information, what is the rationale behind including it in the prompt? Moreover, if the context is omitted, would that not represent a more 'intuitive role' than the one delineated in the study?

- The manuscript would greatly benefit from improvements in its presentation and clarity. For instance, the font size in Figures 3, 4, 7 and 10 is too small to be legible. There are inconsistencies in capitalization, as seen with the word "robust" in the title of Section 4.1, and in Sections 4.4 and 4.5, all the letters in the titles are lowercase. Furthermore, there are formatting issues with the citation styles in the related work section, where "\cite" is used instead of the correct "\citep". These, coupled with numerous grammatical mistakes—including in the abstract—suggest a lack of thorough proofreading and give the impression that the paper may have been hastily composed.

---

> ### Author Response · Authors · 2023-11-22
> **Response to Reviewer WVwf**
>
> Thanks for your comprehensive review. Here are the responses on the questions
>
> **The university of the findings**
>
> Our main contribution lies in the proposed evaluation framework and the findings are also universally applicable. First, we consider the task-solving scenario without preference intervention, which targets the ability of LLMs only. For fairness, we have conducted an instruction selection process (Section 3.3) (Section 3.3) to select instructions from a substantial, non-preferential instruction set within the KRE dataset for each model. This evaluation process ensures the reliability of the two robustness scores —  factual robustness and decision-making style.
>
> Taking the above results as basic scores (original scores), we explore whether can change this internal preference, we implement role-play interventions (a practical method in the real world, where people usually instruct LLMs with specific roles).  This advanced step enables us to measure models’ adaptability in role-playing which is a crucial aspect. Further, we analyze how this inner change affects the factual robustness. As you mentioned there are variances for VR and RR under specific role instruction. It's important to note that these variances do not solely reflect the model's robustness. For example, to some extreme, the model achieves 100 on VR under the intuitive role instruction does not necessarily indicate a model's superior capability to discern incorrect information in the prompt.
>
> In conclusion, the changing scores are based on different scenarios; meanwhile, we want to highlight the importance of such a quantitative evaluation framework for robustness adaptability, and upper bound(Refer to the general response for detailed contributions.).
>
> **For the use of the role play interventions**
>
> Our unique contribution lies in quantitatively assessing the adaptability of these models in such role-playing scenarios and proposing the upper-bound metric using role play instruction. This aspect of adaptability is particularly crucial, given the growing popularity of role play as a method to direct model behavior, yet it has not been quantitatively assessed before.
>
> We have done the experiments when the context is omitted —  the memory assessment (Section 3.2). It is more of an 'intuitive role' indeed. However, it is also meaningful to consider contexts even if instructing asks to overlook them. Taking retrieval augmented LLMs as an example, the retrieval results will be the context and the context can be noisy. In that case, we expect the model's ability to ignore the context. Our proposed evaluation framework is to testify such ability quantitatively, rather than highlighting the role play method itself.

---

> ### Author Response · Authors · 2023-11-23
> **Response to Reviewer WVwf**
>
> Dear reviewer:
>
> Thank you for your careful comments again. Since today marks the final day of discussion, we kindly request once again for your feedback and comments. As shown in your comment "I am willing to raise the score for further explanations.", we sincerely look forward to your further feedback. If you find that our responses have effectively addressed your concerns, can you give us feedback? Your feedback is crucial to us. We apologize for any inconvenience caused.

---

### Official Review · Reviewer_WrsR · 2023-11-04

**Soundness:** 2 fair
**Presentation:** 1 poor
**Contribution:** 2 fair
**Rating:** 3
**Confidence:** 3

**Summary:**

This paper studies the robustness of LLM on two different aspects vulnerable robustness (VR) and resilient robustness (RR). VR measures if the LLM can neglect wrong info in the prompts and respond the correct answers. RR measures if the LLM can respond correctly using the info provided in the prompt. The author proposes a benchmark based on question answering datasets. The author inject some wrong context (i.e., negative contexts) and examples to the prompts in order to measure robustness. They benchmark in several open-sourced and closed-sourced LLMs. The results show that large models are more robust.

**Strengths:**

Robustness of LLM is an important problem. The proposed benchmark seems to be a viable first step towards evaluating the robustness. Another strength is that the study is compreshensive.

**Weaknesses:**

- Notations are confusing.
- What does "high-quality" mean in Section 2?
- What is the definition of commonsense? This term has 14 occurrence in the paper but never been formally defined. This harms the clarity of this paper
-  In Section 4.1, "However, their robustness against negative context introduced by conflicting prompts remains suboptimal. Consequently, as the field progresses, enhancing robustness against adversarial negative context is likely to emerge as a paramount research focus". I agree the result shows that LLM remains suboptimal when negative context are presented. However, I'm not convinced that "adversarial prompts" is a concern since the users have no intention to design adversarial prompts when using LLMs. I can be wrong since how and when adversarial prompts impact LLM users are not specified.
-  In Section 4.1, this sentence is not clear: "This result shows that LLMs prioritize the prompts with factual knowledge more than with commonsense knowledge.". I cannot see why the above result leads to this conclusion. Maybe I don't understand the definition of factual knowledge and commonsense knowledge.
- Figures 2 and 3 are very hard to parse. I suggest splitting them to different subfigures.
- In Section 4.4, "Interestingly, LLaMA the only one that aligns with the Rational Style," From Table 2, it seems that LLaMA is not rational right? Did I read anything wrong?
- What is ”God’s-eye view" instructions?

Overall, the main weakness of this paper is presentation. At its current form, it's very hard to follow and parse the experimental results. Another weakness is that some of analysis are not well-motivated. For example, I don't get the idea of doing decision style analysis.

**Questions:**

- Typo: Section 4.2, "Dose" -> "Does"

---

> ### Author Response · Authors · 2023-11-22
> **Response to Reviewer WrsR (1/2)**
>
> Thanks for your suggestions. Here is our response to each question.
>
> **For the confusion about the notation:**
>
> Can you explain which notations are confusing?
>
> **The meaning of the "high-quality"**
>
> "To ensure high quality, our knowledge robustness evaluation dataset (KRE) dataset extends existing machine reasoning comprehension (MRC) and commonsense reasoning (CR) datasets by automatically generating conflicting cases." Here high quality pointed at the quality of the data sample, including whether the question is rational, and whether the given answer corresponds to the question.
>
> **The definition of commonsense and factual knowledge**
>
> Commonsense knowledge[1], in the context of artificial intelligence, consists of facts about the everyday world that all humans are expected to know. Examples include basic truths like "Lemons are sour" or "Cows say moo". The question about commonsense knowledge requires the interpretation of everyday situations and understanding of the general beliefs about the world.
>
> On the other hand, Factual knowledge refers to information that is based on real, objective facts and evidence. For example the Historical Facts: "The American Civil War lasted from 1861 to 1865." This type of knowledge is often found in encyclopedias, textbooks, and scientific publications. The questions about factual knowledge are focused on seeking specific, evidence-backed information.
>
> In our study, we have selected two Machine Reading Comprehension (MRC) datasets that are based on Wikipedia contexts. This choice is deliberate, as Wikipedia's content which are commonly used as factual knowledge, makes these datasets suitable for testing factual knowledge. Conversely, the chosen Commonsense Reasoning (CR) datasets are inherently designed to test commonsense knowledge.
>
> **Explain for the  "adversarial prompts"**
>
> Here "adversarial prompts" in our paper refer to negative context. That is, we try to introduce a piece of conflicting information in the prompt. We understand that adversarial prompts are well-known in other fields. We will remove it and keep our terms consistent.
>
> **Interpretation for the result " LMs prioritize the prompts with factual knowledge more than with commonsense knowledge"**
>
> Our findings indicate a notable trend (shown in Figure 3): in the MRC dataset, where the input context is factual knowledge, the tested models exhibited higher RR and lower VR compared to RC. A higher RR score indicates that the models benefitted more from the addition of factual knowledge in the MRC task contexts. Conversely, a lower VR score suggests that the models are more influenced by the negative prompts. Thus, we conclude that the baseline models can better utilize factual knowledge than commonsense knowledge from prompt contexts.
>
> **The interpretation of Figures 2 and 3**
>
> We have modified Figures 2, and 3 to make it clearer in our new edition. For Figure 2 (Figure 3 in the revised paper) presents VR and RR together, where the solid bar denotes VR and the dashed bar denotes RR. Both denote more robust models if have longer bars.  We put them together because we can also illustrate the overall robustness — the total length of solid and dashed bars. To make them clearer, we increased the font size of the legend and the label.
>
> Figure 3 (Figure. 4 in the revised paper) explores the impacts of hints in the prompts — if we can improve the model's robustness by giving some hints about the context of the prompt (details in Section 2). First, we can see, from the lines, that neither ChatGPT nor Vicuna showcases any substantial improvements. Second, we use bars to delve into the reason. With prompt hints, although the negative answers (light green bars) decrease, there are more invalid answers (dark green bars), which offsets the improvements.
>
> [1] https://en.wikipedia.org/wiki/Commonsense_knowledge_(artificial_intelligence)

---

> ### Author Response · Authors · 2023-11-22
> **Response to Reviewer WrsR (2/2)**
>
> **For the issue of the  sentence "Interestingly, LLaMA the only one that aligns with the Rational Style"**
>
> It is a writing error,  we have corrected it in the new edition.
>
> **Explaination for the  "God’s-eye view" instructions**
>
> We have deleted the term God-'s eye view instruction in the revised paper for clarity. The instruction is to play an intuitive role when the model has correct memory, and a dependent role when has no correct memory.
>
> **Motivation of doing decision style analysis**
>
> The motivations of decision style analysis are:
>
> 1.To systemically analyze the preferences models enables us to get insights into the models’ behavioral inclinations. Then we can strategically influence and modify these internal preferences to measure their adaptively and thereby exert control over the model's performance.
>
> 2.From a practical perspective, sometimes the conflict between memory and prompts is not related to correctness. That is, both are correct, and it is just about how the models make decisions, e.g., recommendations. Still, this is also important as we want the model's behavior consistent and predictable.
>
> 3.Instructing LLMs to play a specific role is commonly used, while there is no quantitative analysis. We are to fill in this research blank. (For more contributions to this aim, please refer to the general response).

---

> ### Author Response · Authors · 2023-11-23
> **Response to Reviewer WrsR**
>
> Dear reviewer:
> Thank you for your careful comments again. Since today marks the final day of discussion, we kindly request once again for your feedback and comments. As shown in your comment "I am willing to raise the score for further explanations.", we sincerely look forward to your further feedback. If you find that our responses have effectively addressed your concerns, can you give us feedback? Your feedback is crucial to us. We apologize for any inconvenience caused.

---

### Author Response · Authors · 2023-11-22
**General Response: The improved presentation of the paper**

In response to the valuable feedback provided by the reviewers, we have thoroughly revised our manuscript to enhance its presentation and clarity. Here we first summarize our contribution as follows for clarity.

In this work, we propose a systematic framework to quantify the robustness of LLMs in conflict situations where prompts contain conflicting information with the model's memory. Such situations are commonly encountered in real-world applications, particularly within the retrieval augmentation LLM-based product. We assess the robustness of LLMs from two aspects: firstly, an external, objective perspective that evaluates the correctness of the LLMs’ responses, termed "Factual Robustness", and secondly, regardless of the correctness, an internal model perspective that examines the LLMs’ inherent preferences in decision-making, we define as "Decision Style". For reliability, we have tried various instructions in the experiment. We rank the involved model by their factual robustness and unravel their inner preferences.

Further, we deploy a common intervention method "Role Play" to change this preference. This advanced step enables us to measure models’ adaptability in role-playing (e.g., Act as an academic writing assistant). This aspect of adaptability is particularly crucial, given the growing popularity of role play as a method to direct model behavior, yet it has not been quantitatively assessed before. By altering the models’ internal preferences, we explore the effects on their external performance — factual robustness, thereby getting the upper bound of it.  Using these two metrics: adaptability and upper bound. We thus found the reason why some models perform competitively with GPT in standard benchmark while poorly in practice —  their adaptability is way less. That is, role play prompt almost doesn't work.